# A revamped rat reference genome improves the discovery of genetic diversity in laboratory rats

## Graphical abstract

## Authors

Tristan V. de Jong, Yanchao Pan, Pasi Rastas, ..., Robert W. Williams, Jun Z. Li, Hao Chen

## Correspondence

junzli@med.umich.edu (J.Z.L.), hchen@uthsc.edu (H.C.)

## In brief

de Jong et al. evaluated the seventh assembly of the rat reference genome, mRatBN7.2, and found that it reduces base-level errors and increases contiguity, although some misassemblies remain. Gene annotations are now more complete. Analysis of whole genomes representing 120 rat strains/substrains revealed 20 million sequence variations. Phylogenetic analysis refined ancestral relationships among these strains. In addition, a new rat genetic map, along with annotated transcription start sites and alternative polyadenylation sites based on mRatBN7.2, is provided.

## Highlights

- mRatBN7.2 is a rat reference genome with improved contiguity and accuracy

- Gene annotations, from both RefSeq and Ensembl, are improved with mRatBN7.2

- Our analysis of 120 strains/substrains of rats found 20 million sequence variations

- A refined phylogenetic tree reveals the relationships between laboratory rats

 de Jong et al., 2024, Cell Genomics 4, 100527
April 10, 2024 © 2024 The Authors. Published by Elsevier Inc.

## Cell Genomics

CellPress

## Resource

# A revamped rat reference genome improves the discovery of genetic diversity in laboratory rats

Tristan V. de Jong,[1,27] Yanchao Pan,[2,27] Pasi Rastas,[3] Daniel Munro,[4,5] Monika Tutaj,[6,7] Huda Akil,[8] Chris Benner,[9] Denghui Chen,[4] Apurva S. Chitre,[4] William Chow,[10] Vincenza Colonna,[11,12] Clifton L. Dalgard,[13] Wendy M. Demos,[6,7] Peter A. Doris,[14] Erik Garrison,[12] Aron M. Geurts,[6] Hakan M. Gunturkun,[1] Victor Guryev,[15] Thibaut Hourlier,[16] Kerstin Howe,[10] Jun Huang,[1] Ted Kalbfleisch,[17] Panjun Kim,[12] Ling Li,[12,18]

*(Author list continued on next page)*

[1]Department of Pharmacology, Addiction Science, and Toxicology, University of Tennessee Health Science Center, Memphis, TN, USA
[2]Department of Human Genetics, University of Michigan, Ann Arbor, MI, USA
[3]Institute of Biotechnology, University of Helsinki, Helsinki, Finland
[4]Department of Psychiatry, University of California San Diego, San Diego, CA, USA
[5]Department of Integrative Structural and Computational Biology, Scripps Research, San Diego, CA, USA
[6]Department of Physiology, Medical College of Wisconsin, Milwaukee, WI, USA
[7]Rat Genome Database, Medical College of Wisconsin, Milwaukee, WI, USA
[8]Michigan Neuroscience Institute, University of Michigan, Ann Arbor, MI, USA
[9]Department of Medicine, University of California San Diego, San Diego, CA, USA
[10]Tree of Life, Wellcome Sanger Institute, Cambridge, UK

*(Affiliations continued on next page)*

## SUMMARY

The seventh iteration of the reference genome assembly for *Rattus norvegicus*—mRatBN7.2—corrects numerous misplaced segments and reduces base-level errors by approximately 9-fold and increases contiguity by 290-fold compared with its predecessor. Gene annotations are now more complete, improving the mapping precision of genomic, transcriptomic, and proteomics datasets. We jointly analyzed 163 short-read whole-genome sequencing datasets representing 120 laboratory rat strains and substrains using mRatBN7.2. We defined ~20.0 million sequence variations, of which 18,700 are predicted to potentially impact the function of 6,677 genes. We also generated a new rat genetic map from 1,893 heterogeneous stock rats and annotated transcription start sites and alternative polyadenylation sites. The mRatBN7.2 assembly, along with the extensive analysis of genomic variations among rat strains, enhances our understanding of the rat genome, providing researchers with an expanded resource for studies involving rats.

## INTRODUCTION

*Rattus norvegicus* has been used in many fields of study related to human disease.[1] The earliest studies using brown rats appeared in the early 1800s[2,3] The Wistar rats, the ancestor of many laboratory strains, were bred for scientific research in 1906.[4] Over 4,000 inbred, outbred, congenic, mutant, and transgenic strains have been created and are documented in the Rat Genome Database (RGD).[5] Approximately 500 are available from the Rat Resource and Research Center.[6] Several genetic reference populations, including the HXB/BXH[7] and FXLE/LEXF[8] recombinant inbred (RI) families, are also available. Both families, together with 30 diverse classical inbred strains,[9] are now part of the Hybrid Rat Diversity Panel (HRDP), which can be used to quickly generate any of over 10,000 isogenic and replicable F1 hybrids—all of which are now essentially sequenced. The outbred N/NIH heterogeneous stock (HS) rats, derived from eight inbred strains,[10] have been increasingly used for fine mapping of physiological and behavioral traits.[11–15] To date, RGD has annotated nearly 2,400 rat quantitative trait loci (QTLs),[16] mapped using F2 crosses, RI families, and HS rats.

The *Rattus norvegicus* genome was sequenced shortly after the genomes of *Homo sapiens* and *Mus musculus*.[17] The inbred Brown Norway (BN/NHsdMcwi) strain, derived by many generations of sibling matings of stock originally from a pen-bred colony of wild rats,[4] was used to generate the reference. Several updates were released over the following decade.[18–20] Since 2014, most rat genomic and genetic research used the incomplete and problematic Rnor_6.0 assembly.[21,22] mRatBN7.2 was created in 2020 by the Darwin Tree of Life/Vertebrate Genome Project (VGP) as the new genome assembly of the BN/NHsdMcwi rat.[23] The Genome Reference Consortium (GRC) (https://www.ncbi.nih.gov/grc/rat) has adopted mRatBN7.2 as the official rat reference genome.

Here, we report extensive analyses of the improvements in mRatBN7.2 compared with Rnor_6.0. To assist the rat research

**Cell Genomics**

**Resource**

Spencer Mahaffey,[19] Fergal J. Martin,[16] Pejman Mohammadi,[20,21] Ayse Bilge Ozel,[2] Oksana Polesskaya,[4] Michal Pravenec,[22] Pjotr Prins,[12] Jonathan Sebat,[4] Jennifer R. Smith,[6,7] Leah C. Solberg Woods,[23] Boris Tabakoff,[19] Alan Tracey,[10] Marcela Uliano-Silva,[10] Flavia Villani,[12] Hongyang Wang,[24] Burt M. Sharp,[12] Francesca Telese,[4] Zhihua Jiang,[24] Laura Saba,[19] Xusheng Wang,[12,18] Terence D. Murphy,[25] Abraham A. Palmer,[4,26] Anne E. Kwitek,[6,7] Melinda R. Dwinell,[6,7] Robert W. Williams,[12] Jun Z. Li,[2,*] and Hao Chen[1,28,*]

[11]Institute of Genetics and Biophysics, National Research Council, Naples, Italy
[12]Department of Genetics, Genomics and Informatics, University of Tennessee Health Science Center, Memphis, TN, USA
[13]Department of Anatomy, Physiology & Genetics, The American Genome Center, Uniformed Services University of the Health Sciences, Bethesda, MD, USA
[14]The Brown Foundation Institute of Molecular Medicine, Center for Human Genetics, University of Texas Health Science Center, Houston, TX, USA
[15]Genome Structure and Ageing, University of Groningen, UMC, Groningen, the Netherlands
[16]European Molecular Biology Laboratory, European Bioinformatics Institute, Wellcome Genome Campus in Hinxton, Cambridgeshire, UK
[17]Gluck Equine Research Center, Department of Veterinary Science, University of Kentucky, Louisville, KY, USA
[18]Center for Proteomics and Metabolomics, St. Jude Children's Research Hospital, Memphis, TN, USA
[19]Department of Pharmaceutical Sciences, Skaggs School of Pharmacy and Pharmaceutical Sciences, University of Colorado Anschutz Medical Campus, Aurora, CO, USA
[20]Center for Immunity and Immunotherapies, Seattle Children's Research Institute, Seattle, WA, USA
[21]Department of Pediatrics, University of Washington School of Medicine, Seattle, WA, USA
[22]Institute of Physiology, Czech Academy of Sciences, Prague, Czechia
[23]Department of Internal Medicine, Section on Molecular Medicine, Wake Forest University School of Medicine, Winston-Salem, NC, USA
[24]Department of Animal Sciences, Washington State University, Pullman, WA, USA
[25]National Center for Biotechnology Information, National Library of Medicine, National Institutes of Health, Bethesda, MD, USA
[26]Institute for Genomic Medicine, University of California San Diego, La Jolla, CA, USA
[27]These authors contributed equally
[28]Lead contact
*Correspondence: junzli@med.umich.edu (J.Z.L.), hchen@uthsc.edu (H.C.)

community in transitioning to mRatBN7.2, we conducted a broad analysis of a whole-genome sequencing (WGS) dataset of 163 samples from 120 inbred rat strains and substrains. Joint variant calling led to the discovery of 19,987,273 high-quality variants from 15,804,627 sites. Additional resources created during our analysis included a rat genetic map, a comprehensive phylogenetic tree for the 120 (sub)strains, and extensive annotation of identified genes, including their transcription start sites (TSSs) and alternative polyadenylation (APA) sites. This new assembly and its associated resources create a more solid platform for research on the many dimensions of physiology, behavior, and pathobiology of rats and for more reliable and meaningful translation of findings to human populations.

## RESULTS

### High structural and base-level accuracy of mRatBN7.2

All sequencing data for mRatBN7.2 were generated from a male BN rat from the Medical College of Wisconsin (BN/NHsdMcwi, generation F61). The assembly is based on integrating data across multiple technologies, including long-read sequencing (PacBio CLR), 10X linked-read sequencing, BioNano DLS optical map, and Arima HiC. After automated assembly, manual curation corrected most of the apparent discrepancies among data types.[24] While mRatBN7.2 contains an alternative pseudo-haplotype (GCA_015244455.1), we focused our analysis on the primary assembly (GCF_015227675.2).

Over the last six iterations of the rat reference, genome continuity has improved incrementally (Table S1). Contig N50, one measure of assembly quality, has been ~30 kb between rn1 to rn5. Rnor_6.0 was the first assembly to include some long-read data and improved contig N50 to 100.5 kb. mRatBN7.2 further improved N50 to 29.2 Mb (Figure S1). Although this measure of contiguity lags slightly behind the mouse reference genome (contig N50 = 59 Mb in GRCm39, released in 2020) and far behind the first telomere-to-telomere human genome (CHM13) (Table 1), it still marks a large improvement (~290 times higher) over Rnor_6.0 (Figure S2). In another measure, the number of contigs in mRatBN7.2 was reduced by 100-fold compared with Rnor_6.0 and is approaching the quality of GRCh38 for humans and GRCm39 for mice (Table 1).

Although aligning Rnor_6.0 with mRatBN7.2 showed a high level of structural agreement (Figures 1A and S3), we identified 36,500 discordant segments between these two assemblies that are longer than 50 bp (Figure S4). To evaluate these differences, we generated a genetic map (Table S2) using data from 378 families of 1,893 HS rats based on the recombination frequency between 150,835 markers.[15] Comparing the order of the markers and their location on the reference confirmed that the order and orientation of genomic segments are much more accurate in mRatBN7.2. For example, there is a 17.2 Mb inverted segment at proximal Chr 6 between Rnor_6.0 and mRatBN7.2 (Figure 1B). It remains inverted when the genetic distance is plotted with marker location on Rnor_6.0 (Figure 1C) but is resolved when using marker locations on mRatBN7.2 (Figure 1D). This was also true for several other regions (e.g., Figures 1E–1G for Chr 19 and Figures S5 and S6 for all autosomes). These data indicate that most of the segment-wise differences between the two assemblies are due to errors in Rnor_6.0.

**Table 1. Global statistics for the rat, mouse, and human reference genomes**

|  | Rnor_6.0 | mRatBN7.2 | GRCm39 | GRCh38 | CHM13 |
|---|---|---|---|---|---|
| Year published | 2014 | 2021 | 2020 | 2014 | 2021 |
| Total sequence length | 2,870,182,909 | 2,647,915,728 | 2,728,222,451 | 3,209,286,105 | 3,054,832,041 |
| Total ungapped length | 2,729,984,219 | 2,626,580,772 | 2,654,621,837 | 3,049,316,098 | 3,054,832,041 |
| No. of scaffolds | 953 | 176 | 61 | 455 | 24 |
| Scaffold N50 | 145,729,302 | 135,012,528 | 130,530,862 | 145,138,636 | 154,259,566 |
| Scaffold L50 | 8 | 8 | 9 | 9 | 8 |
| No. of contigs | 75,695 | 757 | 347 | 1,431 | 24 |
| Contig N50 | 100,511 | 29,198,295 | 59,462,871 | 56,413,054 | 154,259,566 |
| Contig L50 | 7,346 | 27 | 15 | 19 | 8 |
| Total no. of chromosomes and plasmids | 23 | 23 | 22 | 25 | 24 |

Genomes are downloaded from UCSC Goldenpath. Summary statistics are calculated based on the fasta files of each release using QUAST.[25]

We mapped linked-read WGS data for 36 samples from the HXB/BXH family of strains against both Rnor_6.0 and mRatBN7.2. These included four samples of the reference strain (BN/NHsdMcwi), two samples from the two parental strains—SHR/OlaIpcv and BN-Lx/Cub—and all 30 extent HXB/BXH progeny strains. The mean read depth of the entire HXB dataset, including all parentals, is 105.5× (range 23.4–184.5). From mRatBN7.2 to Rnor_6.0, the fraction of reads mapped to the reference increased by 1%–3% (Figure 2A), and regions of the genome with no coverage decreased by ~2% (Figure 2B). Genetic variants (SNPs and indels) were identified using *Deepvariant*[26] and jointly called for the 36 samples using *GLnexus*.[27] After quality filtering (qual ≥ 30), we identified 8,286,401 SNPs and 3,527,568 indels in Rnor_6.0. Corresponding numbers for mRatBN7.2 are 5,088,144 SNPs and 1,615,870 indels (Figures 2C and 2D). Surprisingly, variants shared by all 36 samples—either homozygous in all samples or heterozygous in all samples—were more abundant when aligned to Rnor_6.0 (1,310,902) than to mRatBN7.2 (143,254) (Figures 2E and 2F). Because we included sequence data from 4 BN/NHsdMcwi rats, including those used for both Rnor_6.0 and mRatBN7.2, the most parsimonious explanation is that these shared variants are due to the wrong nucleotide sequence being recorded as the reference allele in the references. Therefore, mRatBN7.2 reduced base-level errors by 9.2-fold compared with Rnor_6.0.

### Improved gene model annotations in mRatBN7.2

mRatBN7.2-based gene annotations, based on transcriptomic datasets and newly revised multi-species sequence alignments, were released in RefSeq 108 and Ensembl 105. We compared the Rnor6.0 and mRatBN7.2 annotation sets in RefSeq using BUSCO v.4.1.4,[28] focusing on the glires_odb10 dataset of 13,798 genes that are expected to occur in single copy in rodents. Instead of generating a *de novo* annotation with BUSCO using the Augustus or MetaEuk gene predictors, we used proteins from the new annotations, picking one longest protein per gene for analysis. BUSCO reported 98.7% of the glires_odb10 genes as complete in RefSeq 108 (single copy [S] 97.2%, duplicated [D] 1.5, fragmented [F] 0.4%, missing [M] 0.9%). In comparison, analysis of Rnor_6.0 with National Center for Biotechnology Information (NCBI)'s annotation pipeline using the same code and evidence sets showed a slightly higher fraction of fragmented and missing genes and more than double the rate of duplicated genes (S 94.4%, D 3.3%, F 0.9%, M 1.4%). The Ensembl annotation was evaluated using BUSCO v.5.3.2 with the lineage dataset "glires_odb10/2021-02-19." The "complete and single-copy BUSCO" score improved from 93.6% to 95.3%, and the overall score improved from 96.5% in Rnor_6.0 to 97.0% in mRatBN7.2. Both annotation sets demonstrate that mRatBN7.2 is a better foundation for gene annotation, with improved representation of protein-coding genes.

RefSeq (release 108) annotated 42,167 genes, each associated with a unique NCBI GeneID. These included 22,228 protein-coding genes, 7,888 long non-coding RNA (lncRNA), 1,288 small nucleolar RNA (snoRNA), 1,026 small-nuclei RNA (snRNA), and 7,972 pseudogenes. Ensembl (release 107) annotation contained 30,559 genes identified with unique "ENSRNOG" stable IDs. These included 23,096 protein-coding genes, 2,488 lncRNA, 1,706 snoRNA, 1,512 snRNA, and 762 pseudogenes. Although the two transcriptomes have similar numbers of protein-coding genes, RefSeq annotates many more lncRNA (more than 3×) and pseudogenes (almost 10×), and only RefSeq annotates tRNA.

Comparing individual genes across the two annotation sets, Ensembl BioMart reported 23,074 Ensembl genes with an NCBI GeneID, and NCBI Gene reported 24,115 RefSeq genes with an Ensembl ID (Table S3). We note that 2,319 protein-coding genes were annotated with different names (Table S4) by Ensembl and RefSeq. While many of these were caused by the lack of formal gene names in one of the sources, some of them were annotated with distinct names. For example, some widely studied genes, such as *Bcl2*, *Cd4*, *Adrb2*, etc., were not annotated with GeneID in Ensembl BioMart. We found that 18,722 gene symbols were annotated in both RefSeq and Ensembl. Among the 22,247 gene symbols found only in RefSeq, 20,185 were genes without formal names. Further, RefSeq contained a total of 100,958 transcripts, with an average of 2.9 (range: 1–51) transcripts per gene. Ensembl had 54,991 transcripts, with an average of 1.8 (range: 1–10) transcripts per gene (Figure S7).

**Figure 1. mRatBN7.2 corrects structural errors in Rnor_6.0**

(A) Genome-wide comparison between Rnor_6.0 and mRatBN7.2 showed many structural differences, such as a large inversion at proximal Chr 6 and many translocations between chromosomes. Image generated using the NCBI Comparative Genome Viewer. Numbers indicate chromosomes. Green lines indicate sequences in the forward alignment. Blue lines indicate reverse alignment.

(B) The large inversion on proximal Chr 6 is shown in a dot plot between Rnor_6.0 and mRatBN7.2.

(C) A rat genetic map generated using 150,835 binned markers from 1,893 heterogeneous stock rats showed an inversion at proximal Chr 6 between genetic distance and physical distance based on Rnor_6.0, indicating that the inversion is caused by assembly errors in Rnor_6.0.

(D) Marker order and genetic distance from the genetic map on Chr 6 are in agreement with physical distance based on mRatBN7.2, indicating that the misassembly is fixed.

(E–G) Genetic map confirms that many assembly errors on Chr 19 in Rnor_6.0 are fixed in mRatBN7.2.

## mRatBN7.2 improved the mapping of WGS data

We compared mapping results of long-read datasets. Unmapped read numbers declined from 298,422 in Rnor_6.0 to 260,947 in mRatBN7.2, a 12.6% reduction using a PacBio CLR dataset from an SHR rat with a total of 4,508,208 reads. Mapping of Nanopore data from one outbred HS rat (~55× coverage) showed much lower secondary alignment—from ~10 million in Rnor_6.0 to 4 million in mRatBN7.2. Similar to the linked-read

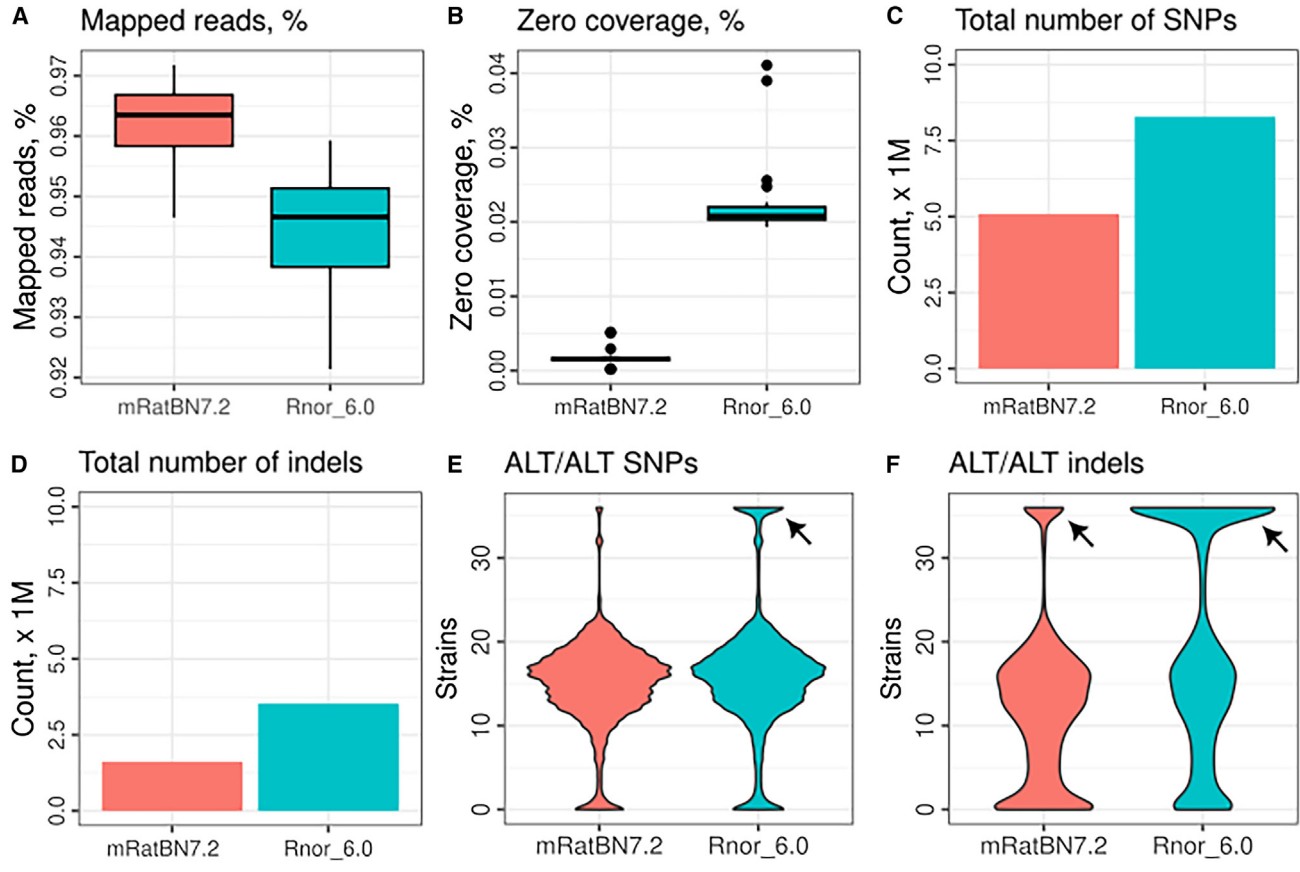

**Figure 2. mRatBN7.2 improved mapping statistics of whole-genome sequencing data**
Summary statistics from mapping 36 HXB/BXH WGS samples against Rnor_6.0 and mRatBN7.2 were compared. Using mRatBN7.2 increased the percentage of reads mapped (A), reduced regions on the reference genome with zero coverage (B), total number of SNPs (C), and indels (D). The presence of a large number of SNPs (E) and indels (F) that are shared by all samples (arrows), including BN/NHsdMcwi, indicates that they are base-level errors in the reference genome.

data, structural variants (SVs) detected in the Nanopore dataset were reduced (Figure S8) using mRatBN7.2.

We also evaluated the effect of reference genome on the identification of SVs. We identified 19,538 unique SVs against Rnor_6.0 in the HXB dataset. In contrast, only 5,458 SVs were found using mRatBN7.2 (see STAR Methods, key resources table, https://doi.org/10.5281/zenodo.10398554). We illustrated these findings by focusing on a small panel of eight samples (Figure S9).

**mRatBN7.2 improved the analysis of transcriptomic and proteomic data**
We analyzed an RNA sequencing (RNA-seq) dataset of 352 HS rat brains (see https://RatGTEx.org) to compare the effect of the reference genome. The fraction of reads aligned to the reference increased from 97.4% in Rnor_6.0 to 98.3% in mRatBN7.2. The average percentage of reads aligned concordantly only once to the reference genome increased from 89.3% to 94.6%. The average percentage of reads aligned to the Ensembl transcriptome increased from 67.7% to 74.8%. Likewise, we examined the alignment of RNA-seq data from ribosomal RNA-depleted total RNA and short RNA in the HRDP. For the total

RNA (>200 bp) samples, alignment to the genome increased from 92.4% to 94.0%, while the percentage of reads aligned concordantly only once to the reference increased from 76.1% to 79.2%. For the short RNA (<200 bp; targeting transcripts 20–50 bp long), genome alignment increased from 95.0% to 96.2% and unique alignment increased from 33.2% to 35.9%. In an snRNA-seq dataset (Figures S10A–S10C), the percentage of reads that mapped confidently to the reference increased from 87.4% on Rnor_6.0 to 91.4% on mRatBN7.2. In contrast, reads mapped with high quality to intergenic regions were reduced from 24.5% on Rnor_6.0 to 10.3% on mRatBN7.2.

We analyzed datasets containing information about transcript start and polyadenylation in a capped short RNA-seq dataset. The rate of unique alignment of TSSs to the reference genome increased by 5% in mRatBN7.2 (Figures S10D and S10E). In this dataset, we identified 42,420 TSSs when using Rnor_6.0 and 44,985 sites when using mRatBN7.2 (see STAR Methods, key resources table, https://doi.org/10.5281/zenodo.10398387). We analyzed 83 whole-transcriptome termini site sequencing[29] datasets using total RNA derived from rat brains and found that 76.97% were mapped against Rnor_6.0, while 80.49% were mapped against mRatBN7.2 (Table S5). We identified 167,136

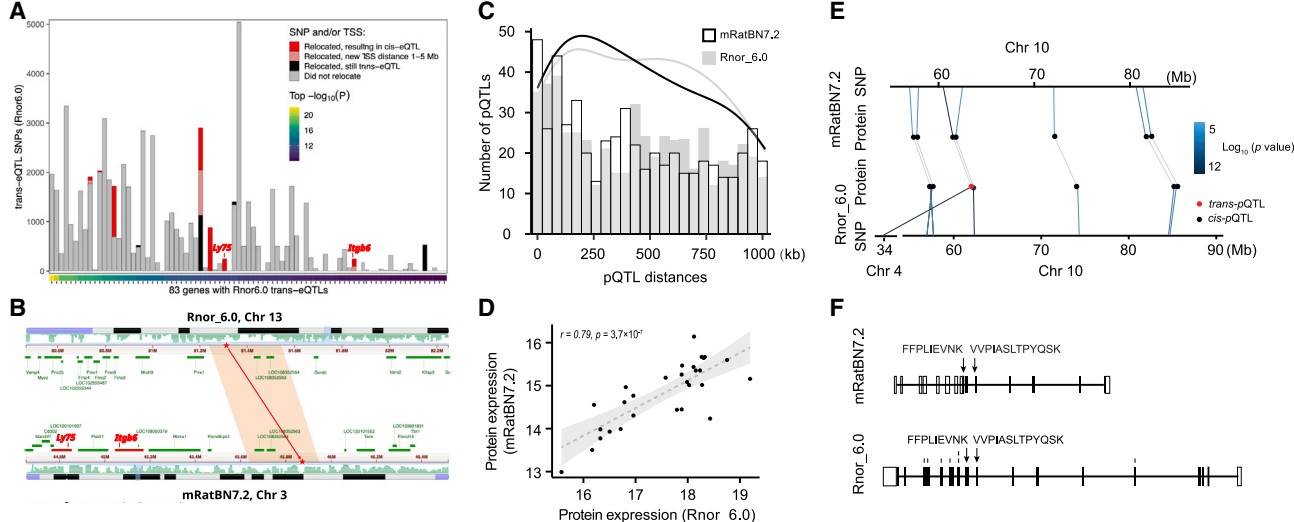

**Figure 3. mRatBN7.2 improves eQTL and proteomic analysis**
Genome misassembly is associated with increased rates of calling spurious *trans*-eQTLs.
(A) Each column represents a gene for which at least one *trans*-eQTL was found at p < 1 × 10⁻⁸ using Rnor_6.0. The color of bars indicate the number of *trans*-eQTL SNP-gene pairs in which the SNP and/or gene transcription start site (TSS) relocated to a different chromosome in mRatBN7.2 and whether the relocation would result in a reclassification to *cis*-eQTL (TSS distance < 1 Mb) or ambiguous (TSS distance is between 1–5 Mb).
(B) Genomic location of one relocated *trans*-eQTL SNP from (A). The SNP is in a segment of Chr 13 in Rnor_6.0 that was relocated to Chr 3 in mRatBN7.2 (red stars), reclassifying the eQTL from *trans*-eQTL to *cis*-eQTL for both *Ly75* and *Itgb6* genes (red bars).
(C) Histogram showing the distance between *cis*-pQTLs and TSS of the corresponding proteins. The distances of pQTLs in mRatBN7.2 tend to be closer than those in Rnor_6.0.
(D) An example of *trans*-pQTL in Rnor_6.0 was detected as a *cis*-pQTL in mRatBN7.2.
(E) Correlation of expression of the protein (the example in B) in Rnor_6.0 and mRatBN7.2.
(F) Different annotations of the exemplar gene in Rnor_6.0 and mRatBN7.2.

APA sites using Rnor_6.0 and 73,124 APA sites using mRatBN7.2 (see STAR Methods, key resources table, https://doi.org/10.5281/zenodo.10398476). For Rnor_6.0, only 76.26% APA sites were assigned to the genomic regions with 18,177 annotated genes (Table S5). In contrast, 81.67% APA sites were mapped to 20,102 annotated genes on mRatBN7.2 (Table S5).

We examined the impact of the upgraded reference assembly on the yield and relative numbers of expression QTL (eQTL) using a large RNA-seq dataset for nucleus accumbens.[30] We identified associations that would be labeled as *trans*-eQTLs using one reference but *cis*-eQTL using the other, due to relocation of the SNP and/or the TSS. The expectation is that assembly errors will give rise to spurious *trans*-eQTLs. We found seven genes associated with one or more strong *trans*-eQTL SNPs using Rnor_6.0 that converted to *cis*-eQTLs using mRatBN7.2 (Figure 3). This constitutes 5.2% (3,261 of 63,148) of the Rnor_6.0 *trans*-eQTL SNP-gene pairs. In contrast, only 0.01% (51 of 404,302) of the Rnor_6.0 *cis*-eQTL SNP-gene pairs became *trans*-eQTLs when using mRatBN7.2. Given the much lower probability of a distant SNP-gene pair remapping to be in close proximity than vice versa under a null model of random relocations, this demonstrates a clear improvement in the accuracy and interpretation of eQTLs when using mRatBN7.2.

To evaluate the effect of reference genome on proteome data, we analyzed a set of data from 29 HXB/BXH strains. At a peptide-identification false discovery rate (FDR) of 1%, we identified and quantified 8,002 unique proteins on Rnor_6.0 compared with 8,406 unique proteins on mRatBN7.2 (5% increase). For protein local expression quantitative trait locus (i.e., *cis*-pQTL), 536 were identified using Rnor_6.0 and 541 were identified using mRatBN7.2 at FDR < 5%. Distances between pQTL peaks and the corresponding gene start site tended to be shorter on mRatBN7.2 than on Rnor_6.0 (Figure 3C). Similar to eQTLs, four proteins with *trans*-pQTL in Rnor_6.0 were converted to *cis*-pQTL using mRatBN7.2. For example, RPA1 protein (Chr10:60,148,794–60,199,949 bp in mRatBN7.2) mapped as a significant *trans*-pQTL (p = 1.71 × 10⁻¹²) on Chr 4 in Rnor_6.0 but as a significant *cis*-pQTL using mRatBN7.2 (p = 4.12 × 10⁻⁶) (Figure 3E). In addition, the expression of RPA1 protein displayed a high correlation between mRatBN7.2 and in Rnor_6.0 (r = 0.79; p = 3.7 × 10⁻⁷) (Figure 3D). Lastly, the annotations for RPA1 were different between the two references (Figure 3F).

## WGS mapping data suggesting potential errors remaining in mRatBN7.2

We analyzed a WGS dataset of 163 inbred rats containing 12 BN samples and 151 non-BN samples, which represented two RI panels and more than 30 inbred strains. After read mapping to mRatBN7.2 and variant calling, we used the anomalies in the genotype data to further assess the quality of mRatBN7.2. This analysis was performed at two levels to detect issues at both the segment level and the individual nucleotide level.

First, we analyzed segment-wise distribution patterns of heterozygous genotypes and no-calls. Since inbred lines are expected

**A**

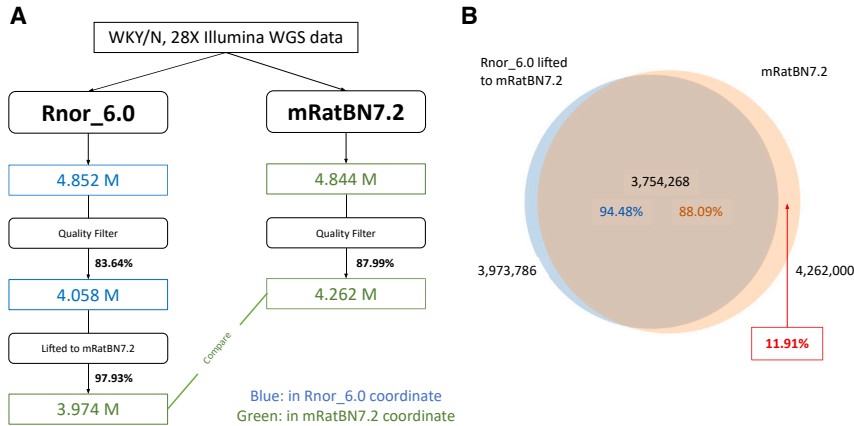

**B**

Rnor_6.0 lifted to mRatBN7.2 mRatBN7.2

3,754,268

94.48% 88.09%

3,973,786 4,262,000

11.91%

**Figure 4. Using WGS data to assess the quality of the Liftover from Rnor_6.0 to mRatBN7.2**
(A) Overview of the workflow using a real WGS sample from a WKY rat. A higher portion of variants passed the quality filter for mRatBN7.2. Among them, 97.93% of the variants were liftable from Rnor_6.0 to mRatBN7.2.
(B) The overlap between variants lifted from Rnor_6.0 and variants obtained by direct mapping sequence data to mRatBN7.2. Approximately 11.9% of the variants that were found from direct mapping were missing from the Liftover.

to show a negligible number of heterozygous sites, genomic regions with an unusually high density of heterozygous genotypes may indicate a segment-wise assembly error. For example, if two segments are tandem repeats of each other and have been "folded" into a single segment in the reference assembly, twice as many reads will map to this region and will produce a high density of heterozygous or multiple-allelic sites. We focused on the 12 BN samples, because BN is the basis of the reference. In all, we identified 673 such "flagged regions," with an average length of 52,199 bp, which collectively covered ~1.4% of the genome (Table S6). This rate is much reduced from that of Rnor_6.0.[22] Furthermore, regions with high heterozygosity were observed among the 151 non-BN samples outside the list of 673 flagged regions. These regions are likely to reflect shared structural variants in the non-BN samples.

We further examined per-site anomalies by identifying sites with Alt/Alt genotypes in most of the 163 strains, including the 12 BN strains. These 129,186 shared variant sites consisted of 33,550 SNPs and 95,636 indels (Table S7). Among them, 117,901 were homozygous alternative genotypes and 11,285 were heterozygous in more than 156 samples (Figure S11). The read depth was 32.2 ± 10.1 for homozygous and 66.3 ± 26.2 for heterozygous variants. Because all samples were inbred, the doubling of read depth for the heterozygous variants strongly suggests that they mapped to regions of the reference genome with collapsed repetitive sequences. This is supported by the location of these variants: homozygous SNPs were more evenly distributed, and heterozygous variants were often clustered in a region (Figure S12). These results indicated that many of the potential errors in mRatBN7.2 are caused by the collapse of repetitive sequences. Functionally, these regions of potential misassembly impact some well-studied genes, such as *Chat*, *Egfr*, *Gabrg2*, and *Grin2a*. After removing the likely errors, we annotated the VCF files resulting from the joint variant calling of 163 samples (see STAR Methods, key resources table, https://doi.org/10.5281/zenodo.10398344) and used these for subsequent analysis.

### Complexities in transitioning from Rnor_6.0 to mRatBN7.2

Liftover tools can convert genomic coordinates between different versions of the reference assembly. However, Liftover can only

return results for regions with one-to-one matches between the two references.

We examined Liftover from Rnor_6.0 to mRatBN7.2 by testing a mock set of variant sites evenly distributed at 1 kb intervals. Of the 2.78 M simulated variants, 92.1% were successfully lifted. Thus, ~8% of the Rnor_6.0 genome does not have a unique match in BN7.2. These "unliftable" sites do not distribute evenly along Rnor_6.0 (Figure S13); rather, they tend to aggregate in regions with no credible match, i.e., lost in BN7.2 (Figure S14).

The rate of "Liftover loss" varies by the type of genomic feature, and as such will be study dependent. We used WGS data for one sample to call variants on Rnor_6.0, then lifted them to mRatBN7.2 and compared the results against the variants called directly on mRatBN7.2. Variants called on mRatBN7.2 had a higher proportion (87.99%) of passing the quality filter than those called on Rnor_6.0 (83.64%), and 97.93% of the variants called on Rnor_6.0 were liftable to mRatBN7.2 (Figure 4A). Among the lifted variant sites, 94.48% had a match in the variant set called directly. However, 11.91% of the variants called on mRatBN7.2 were absent in the lifted variant set (Figure 4B). Thus, complete remapping of the data to the new reference is preferable despite its time and resource costs.

### A comprehensive survey of the genomic landscape of *Rattus norvegicus* based on mRatBN7.2

To facilitate a smooth transition to mRatBN7.2, we conducted a joint analysis of WGS data. We collected WGS for a panel of 163 rats (hereafter referred to as RatCollection, Table S8). RatCollection includes all 30 HXB/BXH strains, 25 FXLE/LEXF strains, and 33 other inbred strains. In total, we covered 88 strains (120 samples at the substrain level)—approximately 80% of the HRDP.

The mean depth of coverage of these samples was 60.4 ± 39.3. Additional sample statistics are provided in Table S9. After removing variants that were potential errors in mRatBN7.2 (above) and filtering for site quality (qual ≥ 30), we identified 19,987,273 variants (12,661,110 SNPs, 3,805,780 insertions, and 3,514,345 deletions) across 15,804,627 variant sites (Table 2). Across the genome, 89.4% of the sites were bi-allelic, and the mean variant density was 5.96 ± 2.20/kb (mean ± SD). The highest variant density of 30.5/kb was found on Chr 4 at 98 Mb (Figure S15). Most (97.9% ± 1.4%) of the variants were

**Table 2. Genetic variants in laboratory populations**

| | Variants sites | Variants | Variant type | | | Predicted impact | | | Alt/Alt homozygous % (mean ± SD) | Genotype qual (mean ± SD) |
|---|---|---|---|---|---|---|---|---|---|---|
| | | | SNPs | Indels | Mixed[a] | High | Low | Moderate | | |
| RatCollection | 15,804,627 | 19,987,273 | 12,661,110 | 7,313,702 | 12,461 | 18,646 | 262,685 | 125,523 | 97.9 ± 1.4 | 70.1 ± 21.2 |
| HS progenitors | 12,418,243 | 16,438,302 | 9,947,112 | 6,479,485 | 11,705 | 6,980 | 205,316 | 94,659 | 98.5 ± 0.3 | 71.4 ± 22.0 |
| FXLE/LEXF[b] | 9,183,562 | 13,070,345 | 7,036,182 | 6,023,391 | 10,772 | 13,988 | 143,623 | 66,186 | 96.8 ± 2.4 | 72.3 ± 21.0 |
| HXB/BXH[b] | 7,520,223 | 11,256,227 | 5,705,592 | 5,541,195 | 9,440 | 10,121 | 115,081 | 53,819 | 97.9 ± 1.2 | 72.2 ± 22.8 |
| SS/SR | 7,171,447 | 10,522,763 | 5,544,220 | 4,969,398 | 9,145 | 8,606 | 111,658 | 51,149 | 98.0 ± 0.9 | 72.3 ± 21.8 |
| LL/LN/LH | 6,923,575 | 10,112,755 | 5,433,556 | 4,670,291 | 8,908 | 4,142 | 111,040 | 52,364 | 98.5 ± 0.3 | 73.2 ± 21.4 |

The RatCollection includes 163 rats (88 strains and 32 substrains, with some biological replicates). The HRDP contains the HXB/BXH and FXLE/LEXF panels as well as 30 or so classic inbreds. Our analysis includes ∼80% of the HRDP. The variants were jointly called using Deepvariant and GLNexus. Variant impact was annotated using SnpEff.
[a]Variants that are combinations of insertions, deletions, or SNPs.
[b]Including parental strains.

homozygous at the sample level, confirming the inbred nature of most strains, with a few exceptions (Figure S16).

To analyze the phylogenetic relationships of these 120 strains/substrains, we created an identity-by-state (IBS) matrix using 11,585,238 high-quality bi-allelic SNPs (Table S10). Distance-based phylogenetic trees of all strains and substrains are shown in Figure 5A. The mean IBS for classic inbred strains was 72.30% ± 2.35%, where BN has the smallest IBS (64.76%).

This phylogenetic tree included all major populations of laboratory rats, such as the eight inbred progenitor strains of the outbred HS rats (ACI/N, BN/SsN, BUF/N, F344/N, M520/N, MR/N, WKY/N, and WN/N).[10] The number of sites with a non-reference allele in each of these strains is shown in Figure 5B; WKY/N contributed the largest number of non-reference alleles (Figure 5C). The distribution of the variant sites from all eight strains across the chromosomes is shown in Figure 5D. Collectively, these 8 strains accounted for 78.6% of all non-reference alleles in the RatCollection. Conversely, 141,556 of these variant sites were not found in any other strains. Although none of these founder strains are alive today, based on IBS, we identified 6 living proxies of the HS progenitors that were over 99.5% similar to the original progenitor strains: ACI/EurMcwi, BN/NHsdMcwi, F344/DuCrl, M520/NRrrcMcwi, MR/NRrrc, and WKY/NHsd (Table S11). The best matches of the remaining strains were much less similar: BUF/Mna was the best approximation (73.6%) to BUF/N, and WAG/RijCrl was the closest (72.0%) to WN/N.

Our phylogenetic tree also included two families of RI rats. The HXB/BXH family was generated from SHR/OlaIpcv and BN-Lx/Cub. Together with their parental strains, we identified 7,520,223 variant sites in this population (Table 2). Approximately 24.1%–53.5% of the alleles at these sites of each RI strain were derived from SHR/OlaIpcv. While the majority of the strains were highly inbred, with close to 98% of the variants being homozygous (Figure S17), one exception was BXH2, in which 7.7% of variants were heterozygous (Figure S16), likely due to a recent breeding error. The LEXF/FXLE RI family was generated from LE/Stm and F344/Stm. We discovered 9,183,562 variant sites from the parental strains and 25 strains of this family (Table 2). We expect the final variant count of the LEXF/FXLE panel

to be similar because both parental strains are included in our analysis. The overall rate of homozygous variants in the FXLE/LEXF family (96.8% ± 2.4%) was lower than in other inbred rats (Figure S16). In particular, 15.6% of the sites from the FXLE24 samples are heterozygous, indicating likely breeding errors.

In addition, our analysis also included sets of two or three strains generated by selective breeding for certain traits, such as the Dahl salt-snsitive (SS) and Dahl salt-resistant (SR) strains for studying hypertension. These two strains contain 7,171,447 variant sites compared with mRatBN7.2 (Table 2), with 1,024,283 variants unique to SR and 920,234 variants unique to SS. These strain-specific variants were found throughout the genome (Figure S18). A similar pattern was found for the Lyon hypertensive (LH), hypotensive (LL), and normotensive (LN) rats (Table 2) selected from outbred Sprague-Dawley rats for studying blood pressure regulation.[31] Only 281,972, 289,112, and 262,574 variants were unique to LH, LL, and LN, respectively. In agreement with a prior report,[32] these variants were clustered in a handful of genomic hotspots (Figure S19).

### Impact of rat variants on genetic studies of human diseases

We used SnpEff (v.5.0e) to predict the impact of the 19,987,273 variants in the RatCollection based on RefSeq annotation. Among these, 18,646 variants near coding genes were predicted to have a high impact (i.e., causing protein truncation, loss of function, or triggering nonsense-mediated decay, etc.) on 6,667 genes, including 3,930 protein-coding genes (Table S12).

Among the predicted high-impact variants, annotation by RGD disease ontology identified 2,601 variants affecting 2,079 genes that were associated with 3,261 distinct disease terms. Cancer/tumor (878), psychiatric (612), intellectual disability (505), epilepsy (319), and cardiovascular (304) comprised the top 5 disease terms, with many genes associated with more than one disease term. A mosaic representation of the number of high-impact variants per disease term for each strain is shown in (Figure S20). The top disease categories for the two RI panels are comparable, including cancer, psychiatric disorders, and cardiovascular disease. The disease ontology annotations for

**A**

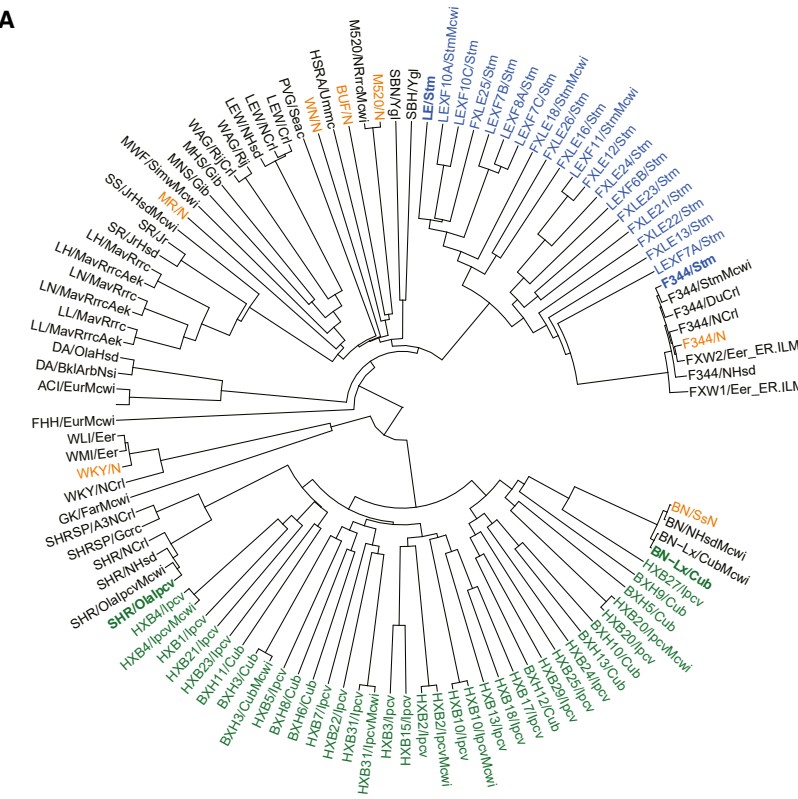

**Figure 5. Phylogenetic relationship and genetic diversity of 120 strain/substrains of laboratory rats**

(A) The phylogenetic tree was constructed using 11.6 million biallelic SNPs from 163 samples. Strains/substrains with duplicated samples were condensed. Strains highlighted with bold fonts are parental strains for RI panels. Green, HXB/BXH RI panel; blue, FXLE/LEXF RI panel; orange, progenitors of the HS outbred population.

(B) The HS progenitors contain 16,438,302 variants (i.e., 82.2% of the variants in our collection of 120 strain/substrains) based on analysis using mRatBN7.2. Among these, 10,895 are shared by all eight progenitor strains. The number of variants that are unique to each specific founder is noted.

(C) The total number of variants per strain, with the total number unique to each strain marked.

(D) The number of variants shared across strains per chromosome.

**B**

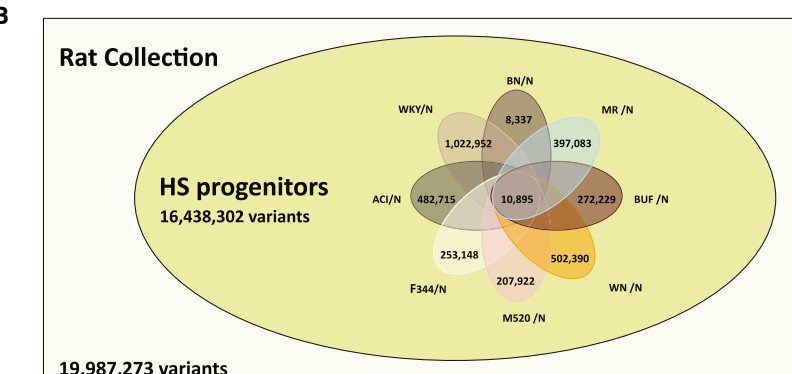

**C** **D**

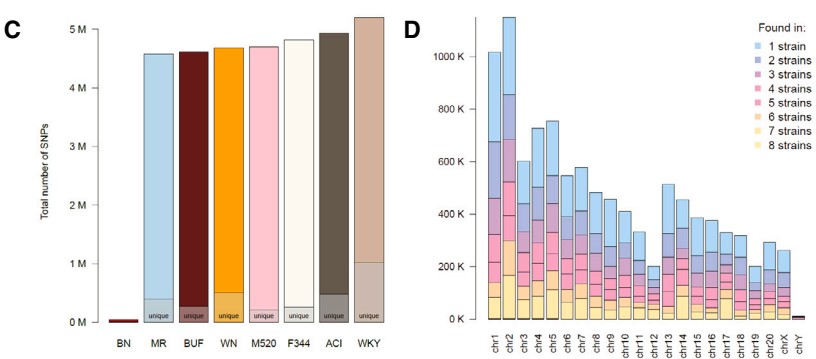

variants in the HXB/BXH, FXLE/LEXF RI panels, and SS/SR, as well as LL/LN/LH selective bred strains, are summarized in Figures S21 and S22 and Tables S13 and S14.

We further annotated genes using the human genome-wide association studies (GWASs) catalog.[33] Among the rat genes with high-impact variants, 2,034 have human orthologs with genome-wide significant hits associated with 1,393 mapped traits (Table S15). The most frequent variant type among these rat genes was frameshift (1,557 genes), followed by splice donor variant (136 genes) and gain of stop codon (116 genes). Although rats and humans do not share the same variants, strain with these variants can potentially be a useful model for related human diseases at the gene level.

## DISCUSSION

We systematically evaluated mRatBN7.2 and confirmed that it improved assembly quality over its predecessor, Rnor_6.0. As a result, mRatBN7.2 improved the analysis of omics datasets and now enables powerful and efficient genome-to-phenome analysis of 20 million variants segregating in the laboratory rat.

The improvements in mRatBN7.2 are based on new assembly methods and long-read data.[23] However, the PacBio CLR reads used in mRatBN7.2 have lower base accuracy than Illumina short reads.[34,35] Although Illumina data were used to polish the assembly,[23] polishing methods do not correct all base-level errors.[34,36] Our joint analysis of 163 WGS datasets identified 129,186 sites with likely nucleotide errors in mRatBN7.2. Another source of potential error is the assembly method itself. mRatBN7.2 was assembled with VGP pipeline v.1.6,[23,37] which assembles the long reads into a diploid genome containing a primary and an alternative assembly.[38] This pipeline is well suited to assemble diploid genomes. When applied to BN/NHsdMcwi, a fully inbred rat, the pipeline classified some duplications with small variants as two haploids, resulting in a collapsed repeat.[39] A telomere-to-telomere assembly[40] is needed to fully resolve these issues.

Liftover enables direct translation of genomic coordinates between different references, thereby reducing the cost of transitioning to a new reference. We found that 92.05% of simulated variants and 97.93% of variants from a WKY/N sample identified on Rnor_6.0 were lifted successfully to mRatBN7.2. This difference is attributable to the large number of simulated variants located in regions of low complexity. While promising, we found that reanalysis of the original sequencing data using mRatBN7.2 discovered an additional 507,700 variant sites, located primarily in regions not present in Rnor_6.0. Thus, remapping to mRatBN7.2 is preferred over using Liftover even though the latter is more convenient.

To facilitate the transition for studies of specific rat models, we mapped WGS data of 88 strains (120 substrains and 163 samples) to mRatBN7.2. Joint analysis identified 20.0 million variants from 15.8 million sites. This analysis expanded many prior analyses of rat genomes. For example, Baud et al.[11] analyzed 8 strains against Rnor3.4 and found 7.9 million variants, Atanur et al.[41] analyzed 27 rat strains against Rnor3.4 and reported 13.1 million variants, and Hermsen et al.[42] analyzed 40 strains against Rnor5.0 and reported 12.2 million variants. Most recently, Ramdas et al.[22] analyzed 8 strains and reported 16.4 million variants. In addition to

using the latest reference genome and an expanded number of strains, our pipeline depends on Deepvariant and GLNexus, which have been shown to improve call set quality, especially on indels.[27,43] Thus, our data provide the most comprehensive analysis of genetic variants in the laboratory rat population to date and, of comparative interest, is twice the variant count of the highly used mouse BXD family.[44,45]

This collection of rats represented a wide variety of rats that are used in genetics studies today, such as the full HXB/BXH panel and 27 strains/substrains of the FXLE/LEXF RI panel. Together with the inbred strains, they cover about 80% of the rats in the HRDP.[46] The new variant set from our analysis provide a much-needed boost in mapping precision.[8,46,47]

The outbred N/NIH HS rats are another widely used rat genetic mapping population.[13] Our analysis is in agreement with Ramdas et al.[22] in identifying 16 million variants in the progenitors. Updated genetic data will be useful in HS genetic mapping studies, such as for the imputation of variants. Although live colonies of the original NIH HS progenitor strains are no longer available, we identified six living substrains that are genetically almost identical to the original progenitors (99.5% IBS), while the closest match to the other two strains has IBS of approximately 70% (Table S11). These living substrains will be useful in many ways, such as studying the effect of variants identified in genetic mapping studies. Our analysis also included groups of inbred rats segregated by less than 2 million variants. For example, the LH/LN/LL family,[31,32] the SS/SR family (Table S14), as well as several pairs of near isogenic lines with distinctive phenotypes. These rats can be exploited to identify causal variants using reduced complexity crosses.[48] Updated genotyping data for these populations will benefit all ongoing studies that use these rats.

A trove of genetic associations has been established using laboratory rats. For example, studies using RI panels have identified loci that affect cardiovascular disease,[49] hypertension,[50,51] diabetes,[52] immunity,[52] tumorigenesis,[53] and tissue-specific gene expression profiles.[54] Using the outbred HS rats, Baud et al.[11] reported 355 QTLs associated with phenotypes representing six diseases. Other studies using the HS population have revealed genetic control of behavior,[15,55] obesity,[56] and metabolic phenotypes.[57] Many of these data are available from the RGD.[16] While all these data were analyzed using previous versions of the reference genome, reanalyzing them using updated genomic data could lead to novel discoveries.[58]

By generating F1 hybrids from these inbred lines with sequenced genomes, novel phenotypes can be mapped onto completely defined genomes: the 82 sequenced HRDP strains can produce any of 6,642 isogenic but entirely replicable F1 hybrids with completely defined genomes. Studies of these "sequenced" F1 hybrids avoid the homozygosity of the parental HRDP strains and enable a new phase of genome-phenome mapping and prediction.[44] While several genetic mapping studies using the HRDP are currently underway, the large number of variants in the HRDP (cf., the hybrid mouse diversity panel contains about 4 million SNPs[59]) and WGS-based genotype data will further encourage the use of this resource.

We identified 18,646 variations likely to have a high impact on 6,667 genes and many more variants predicted to have regulatory

effects on gene expression (Table S13). Some of these variants are supported by the literature. For example, the SS rats develop renal lesions with hypertension and have high impact mutations in *Capn1*[60] and *Procr*,[61] both associated with kidney injuries, as well as in *Klk1c12*, associated with hypertension.[62] These annotations identified many genes with variants that either result in a gain of STOP codon or cause a frameshift, which could lead to the loss of function. Strains harboring these variants can be explored to investigate the function of these genes.[63]

These functional annotations are highly relevant when the human ortholog of these genes is associated with certain traits in human GWASs (Table S15). For example, *CDHR3* is associated with smoking cessation.[64] A gain of STOP mutation in the *Cdhr3* gene was found in only one parent of both RI panels (SHR/Olalpcv and LE/Stm). Using these RI panels to study the reinstatement of nicotine self-administration, a model for smoking cessation, will likely provide insights into the role of *CDHR3* in this behavior. Furthermore, a near-complete catalog of strain-specific alleles and functional prediction provide a stable source of potential candidate variants for interpreting genetic mapping results.

Our phylogenetic analysis agrees with those published previously[41,65] and confirmed that BN/NHsdMcwi is an outgroup to all other common laboratory rats (Figure 5A). This is consistent with its derivation from a pen-bred colony of wild-caught rats[4] and previous studies that included wild rats.[66] Thus, mapping sequence data from other strains to the BN reference yields a greater number of variants (but at slightly reduced mapping quality) than using a hypothetical reference that is genetically closer to the commonly used laboratory strains. This so-called reference bias has been observed in genomic[67] and transcriptomic data analyses.[68] It should be noted that no individual strain is a perfect representation of a population. Instead, the nascent field of pangenomics,[69] where the genomes of all strains can be directly compared with each other, provides a promising future in which all variants can be compared between individuals directly without the use of a single reference genome. While this pangenomic approach can be applied to short-read data,[70] it will be especially powerful when individual genomes are all assembled from long-read sequence data, some of which are already available.[71] This approach will enable a complete catalog of all genomic variants, including SVs and repeats, that differ between individuals.

Additional rat genomic resources we generated include a new rat genetic map with 150,835 markers. We envision that this map will have multiple applications. For example, this map can be integrated into the *de novo* assembly process, as demonstrated by the new assembly of the stickleback genome.[72,73] Unlike human and mouse genomes,[74] functional elements in the rat remain poorly annotated. Our analysis produced a list of TSS locations for genes expressed in the prefrontal cortex or nucleus accumbens and APA sites of genes expressed in the brain. These new datasets will further the study of gene regulatory mechanisms in rats.

Research using the *Rattus norvegicus* has made important contributions to the understanding of human physiology and diseases. An updated reference genome provides not only a valuable resource for future studies but also opportunities for analyzing existing data to gain new insights. The rich literature

on laboratory rats, combined with the complex genomic landscape revealed in our survey, demonstrates that the rat is an excellent model organism for the next chapter of biomedical research.

## Limitations of the study

While our comparative analyses reported many improvements in mRatBN7.2 over Rnor_6.0, we also found that mRatBN7.2 contains a few hundred segments with potential disassembly, as well as base-level errors. These segments affect gene annotation and the accuracy and completeness of Liftover. While we are confident in the informatic analysis of these segments, we did not empirically validate these findings. Future iterations of the rat reference genome need to reach telomere-to-telomere completion and eventually be expanded to a multi-strain pangenome reference.

## STAR★METHODS

Detailed methods are provided in the online version of this paper and include the following:

- KEY RESOURCES TABLE
- RESOURCE AVAILABILITY
  - Lead contact
  - Materials availability
  - Data and code availability
- EXPERIMENTAL MODEL AND STUDY PARTICIPANT DETAILS
- METHOD DETAILS
  - Calculating genome assembly statistics
  - Analysis of WGS data
  - Sample quality control
  - Identification of genomic regions with potential mis-assembly in mRatBN7.2
  - Evaluating Liftover
  - Identify live strains that are close to HS progenitors
  - Constructing a genetic map using genetic data from a large HS cohort
  - Phylogenetic tree
  - RNA-seq data
  - eQTL relocation analysis
  - Capped small (cs)RNA-seq data
  - Single nuclei (sn) RNA-seq data
  - Transcriptome termini site sequencing
  - Brain proteome data
  - Identifying potentially mislabeled samples
  - Ensembl annotation
  - RefSeq annotation

## SUPPLEMENTAL INFORMATION

## ACKNOWLEDGMENTS

This work is supported by the Academy of Finland (grant no. 343656) to P.R.; NIH NHLBI R01HL064541 and P01HL149620 and Office of the Director

R24OD024617 to M.T.; NIH NIDA U01DA043098 to H.A.; NIH NIDA U01DA051972 to C.B.; NIH NHLBI R01HL064541 and Office of the Director R24OD024617 to W.M.D.; NIH NHLBI P01HL149620 and Office of the Director R24OD024617 to A.M.G.; Wellcome Trust WT222155/Z/20/Z to T.H.; NIH R01HG011252 to T.K.; Wellcome Trust WT222155/Z/20/Z to F.J.M.; NIH R01GM140287 to P.M.; a program from the National Institute for Research of Metabolic and Cardiovascular Diseases (Program EXCELES, ID project no. LX22NPO5104) funded by the European Union – Next Generation EU to M.P.; National Institute of Food and Agriculture, United States Department of Agriculture (2016-67015-24470/2020-67015-31733/2022-51300-38058/2023-67015-39566/2023-67015-40080) to Z.J.; NIH NIDA U01DA051234 to J.S.; NIH NHLBI R01HL064541, NHGRI U24HG010859, and Office of the Director R24OD024617 to J.R.S.; NIH NIDA P50DA037844 to L.C.S.W. (HS rats); NIH NIAAA R24AA013162 to B.T.; NIH NIDA U01DA050239 and U01DA051972 to F.T.; NIH NIDA P30DA044223 to L.S.; the National Center for Biotechnology Information of the National Library of Medicine (NLM), National Institutes of Health to T.D.M.; NIH NHLBI R01HL064541 and P01HL149620, NHGRI U24HG010859, and Office of the Director R24OD024617 to A.E.K.; NIH Office of the Director grant R24OD024617 to M.R.D. (HRDP); NIH NIDA U01DA047638 and P30DA044223 to R.W.W.; NIH NIDA U01DA043098 to J.Z.L.; and NIH NIDA U01DA047638, P50DA037844, and R01DA048017 to H.C. The majority of the computation for this work was performed on the University of Tennessee Infrastructure for Scientific Applications and Advanced Computing (ISAAC) computational resources.

This work is dedicated to the memory of Dr. Mary Shimoyama.

## AUTHOR CONTRIBUTIONS

Conceptualization, A.A.P., R.W.W., J.Z.L., and H.C.; formal analysis, T.V.d.J., Y.P., P.R., D.M., M.T., X.W., T.D.M., J.Z.L., and H.C.; visualization, T.V.d.J., Y.P., P.R., D.M., F.T., Z.J., L.S., X.W., and J.Z.L.; investigation, H.A., C.B., D.C., A.S.C., W.C., V.C., W.M.D., P.A.D., E.G., A.M.G., H.M.G., V.G., T.H., K.H., J.H., T.K., P.K., L.L., S.M., F.J.M., P.M., A.B.O., O.P., P.P., J.S., J.R.S., L.C.S.W., B.T., A.T., M.U.-S., F.V., H.W., F.T., Z.J., and L.S.; resources, C.L.D., M.P., A.E.K., and M.R.D.; data curation, W.M.D., A.M.G., J.R.S., A.E.K., and M.R.D.; writing – original draft, T.V.d.J., Y.P., P.R., D.M., T.H., L.S., X.W., T.D.M., J.Z.L., and H.C.; writing – review & editing, A.M.G., B.M.S., L.S., X.W., T.D.M., A.A.P., A.E.K., M.R.D., R.W.W., J.Z.L., and H.C.; supervision, J.Z.L. and H.C.

## DECLARATION OF INTERESTS

The authors declare no competing interests.

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

**Cell Genomics**
**Resource**

## STAR★METHODS

### KEY RESOURCES TABLE

| REAGENT or RESOURCE | SOURCE | IDENTIFIER |
|---|---|---|
| **Deposited data** | | |
| Structural variants in HXB RI panel (VCF) | This paper | https://doi.org/10.5281/zenodo.10398554 |
| Join calling of 163 WGS rat samples (VCF) | This paper | https://doi.org/10.5281/zenodo.10398344 |
| Transcription start site for 2 brain regions | This paper | https://doi.org/10.5281/zenodo.10398387 |
| Alternative polyadenylation sites | This paper | https://doi.org/10.5281/zenodo.10398476 |
| Potential misassembled regions | This paper | https://doi.org/10.5281/zenodo.10428255 |
| Chromosomal dot plots between Rnor_6.0 and mRatBN7.2 | This paper | https://doi.org/10.5281/zenodo.10515796 |
| The order of genetic markers and the distances from a rat genetic map compared to their locations in Rnor_6.0 | This paper | https://doi.org/10.5281/zenodo.10552387 |
| The order of genetic markers and the distances from a rat genetic map compared to their locations in mRatBN7.2 | This paper | https://doi.org/10.5281/zenodo.10552453 |
| Code and data for phylogenetic analysis | This paper | https://doi.org/10.5281/zenodo.10520242 |
| nucleus accumbens core eQTL dataset | Munro et al.[30] | NAcc, 75 samples |
| csRNA-seq data | Duttke et al.[75] | N/A |
| Annotation of the mRatBN7.2 assembly | O'Leary et al.[76] | Release 108 |
| Code and computational notebook | This paper | https://doi.org/10.5281/zenodo.10552887 |
| **Experimental models: Organisms/strains** | | |
| Sample metadata for RatCollection | This paper | https://doi.org/10.5281/zenodo.10552790 |
| **Software and algorithms** | | |
| QUAST | Mikheenko et al.[25] | N/A |
| paftools | Kalikar et al.[77] | N/A |
| BWA mem | Li and Durbin[78] | N/A |
| GATK | Poplin et al.[79] | N/A |
| LongRanger | | N/A |
| Minimap2 | Li[80] | N/A |
| Deepvariant | Poplin et al.[26] | N/A |
| GLNexus | Yun et al.[27] | N/A |
| SURVIVOR | Jeffares et al.[81] | N/A |
| SnpEff | Cingolani et al.[82] | N/A |
| RepeatMasker | Tarailo-Graovac and Chen[83] | N/A |
| GeneCup | Gunturkun et al.[84] | N/A |
| UCSC Liftover tool | Hinrichs et al.[85] | N/A |
| PLINK | Purcell et al.[86] | N/A |
| Lep-MAP3 | Rastas[87] | N/A |
| bcftools | Danecek et al.[88] | N/A |
| MEGA | Tamura et al.[89] | N/A |
| ggtree | Yu et al.[90] | N/A |
| STAR aligner | Dobin et al.[91] | N/A |
| HOMER's findPeaks tool | Duttke et al.[92] | N/A |
| QTLtools | Delaneau et al.[93] | N/A |
| UniProt | UniProt Consortium[94] | N/A |

## RESOURCE AVAILABILITY

### Lead contact

Further information and requests for resources should be directed to and will be fulfilled by the lead contact, Hao Chen (hchen@uthsc.edu).

### Materials availability

The Hybrid Rat Diversity Panel (HRDP) consists of 96 inbred strains, and includes 30 classic inbred strains and both of the large RI families discussed in the prior sections (BXH/HXB and LEXF/FXLE). The panel is being cryo-resuscitated and cryopreserved at the Medical College of Wisconsin for use by the scientific community.

### Data and code availability

- The RatCollection contains 163 WGS samples from 88 strains and 32 substrains. It includes new data from 128 rats (key resources table) and 36 datasets downloaded from NIH SRA. The detailed sample metadata are provided in the Table S8. A total of 82 members of the HRDP have been sequenced using Illumina short-read technology, including all 30 extant HXB strains, 23 of 27 FXLE/LEXF strains, and 25 of 30 classic inbred strains. RatCollection includes all 30 strains of the HXB/BXH family, 27 strains in the FXLE/LEXF family, and 33 other inbred strains. In total, we covered 88 strains and 32 substrains. It contains approximately 80% of the HRDP. WGS data generated for this work have been uploaded to NIH SRA (see Table S8 for SRA IDs). Other resources are available from Zenodo (See key resources table). The code for the custom R, Python and Bash scripts for data analysis is available from a github repository: https://github.com/hanyoupan/rat-manuscript (https://doi.org/10.5281/zenodo.10552887)
- Any additional information required to reanalyze the data reported in this paper is available from the lead contact upon request.

## EXPERIMENTAL MODEL AND STUDY PARTICIPANT DETAILS

All rats used for whole genome sequencing are naive adult males. Their strain and substrain identification as well as RRIDs are provided in the Table S8.

## METHOD DETAILS

### Calculating genome assembly statistics

We obtained all assemblies from UCSC Goldenpath, with the exception of CHM13_T2T_v1.1, which was downloaded from the T2T Consortium GitHub page. We used QUAST[25] to calculate common assembly metrics, such as contig and scaffold N50, using a consistent standard across all assemblies. We defined each entry in the fasta file as a scaffold, breaking them into contigs based on continuous Ns of 10 or more. No scaffolds below a certain length were excluded from the analysis.[23] Our scripts and intermediate results can be found in the supplementary. Nx plots were generated using a custom Python script and the fasta files as inputs. Structural differences between Rnor_6.0 and mRatBN7.2 were evaluated using paftools.[77]

### Analysis of WGS data

We used different methods for mapping data, based on the sequencing technologies. For illumina short read data, fastq files were mapped to the reference genome (either Rnor_6.0 or mRatBN7.2) using BWA mem.[78] GATK[79] was then used to mark PCR duplicates in the bam files. For 10x chromium linked reads, fastq files were mapped against the reference genomes using LongRanger (version 2.2.2). Long read data were mapped using Minimap2,[80] Deepvariant[26] (ver 1.0.0) was then used to call variants for each sample. Joint calling of variants for all the samples was conducted using GLNexus.[27] Large SVs detected by LongRanger were merged using SURVIVOR[81] per reference genome. SVs detected in less than two samples were removed. Variants with the QUAL score less than 30 were removed. The impact of variants were predicted using SnpEff,[82] using RefSeq annotations. Overlap between SNPs and repetitive regions were identified with RepeatMasker.[83] Disease ontology was retrieved from the Rat Genome Database.[16] Disease associations were related to nearest genes as predicted by SnpEff. Subsequent analyses were conducted using custom scripts in R or bash. Circular plots were generated with the Circos plots package in R. Functional consequences of variants on genes were searched in PubMed via GeneCup.[84]

### Sample quality control

To ensure the quality of data from 168 rats, we conducted a thorough examination of the missing call rate and read depth per sample. We determined that a per sample missing rate of 4% and average read depth of 10 were appropriate based on the data distribution. We identified and removed 4 samples with high missing rates (SHR/NCrlPrin_BT.ILM, WKY/NHsd_TA.ILM, GK/Ox_TA.ILM, FHL/EurMcwi_TA.ILM) and 2 samples with low read depth (WKY/NHsd_TA.ILM, BBDP/Wor_TA.ILM), resulting in a total of 5 samples removed.

### Identification of genomic regions with potential mis-assembly in mRatBN7.2

We used WGS-derived genotype data for 163 samples to detect genomic regions with unusually high densities of heterozygous genotypes (i.e., the "high-het" regions). Since the samples are from inbred animals, high-het regions could arise from tandemly repeated segments in the BN genome "folded" into a single region in mRatBN7.2. While read-depth data represent an independent source of information, here we focused on the distribution patterns of heterogeneous genotypes in the 12 BN animals. Indeed, along the genome, the per-site heterozygote counts, ranging 0 to 12 across the 12 BN samples, tend to be high in certain discrete regions, which also tend to have high counts of NA (the "no-calls"). We used the het+NA counts as the scanning statistics for segment-wise switches between a high-state and a low-state, akin to using arrayCGH intensity data or sequencing read depths to detect DNA copy-number variants (CNVs). Specifically, we added 2 to the per-site het+NA counts, to mimic the situation with 2 DNA copies in baseline regions and up to 14 copies in the high-regions. The logged values, ranging from log2(2) to log2(14), are "segmented" by using the segment command in R package "DNAcopy," with parameters min.width = 5,alpha = 0.001. The results tend to be over-segmented, and they are merged by a custom R script with the following merging rules. Starting from the first three segments, we merge the first and the second segments if one of the four conditions are met: (1) both have mean values above 1.5, which is an empirically derived threshold in our log2(x+2) scale, (2) both have mean values below 1.5, (3) the three segments are high-low-high but the middle segment is less than 5 kb, hence a "positive flicker," or (4) the three segments are low-high-low with the middle segment shorter than 5 kb: a "negative flicker." If the first two segments are merged, the next "triplet" to be evaluated are the merged 1–2, the previous #3 segment, and the newly called-up $4 segment. If no merging is executed, the scan moves by one segment to evaluate the next "triplet": #2–4. This operation was repeated until we reached the end of the chromosome. Altering the merging parameters will yield a somewhat different set of "flagged" regions. The current merged list contains 673 high-segments over the 20 autosomes, covering about 1.4% of mRatBN7.2 and having an average length of 52,199 nt. Future iterations will incorporate distribution patterns of read depth, multi-allelic sites and, if possible, linked read information. Similar scans can be performed for the 151 non-BN samples. Our preliminary data show that most of the regions flagged by the 12 BN samples also show high Het+NA counts in non-BN samples. In addition, some other heterozygous genotypes in non-BN samples fall outside the flagged regions, either individually or in new segments not flagged in BN samples. Many of them likely represent single-site or segment-wise differences between these strains and the reference strain: BN, and they may vary in a strain-specific fashion.

### Evaluating Liftover

The Liftover tool is evaluated using both simulated and real data. The simulated data is an evenly 1000-base pair-spaced bed file covering Rnor_6.0 (2,782,023 sites within the bed file). The simulated dataset is used to study what portion of the genome is liftable, and the distribution of the unliftable and liftable sites. The real dataset is used to evaluate the accuracy and utility of the Liftover process, we compared the variants obtained through the Liftover process to those directly called from the data. We mapped WGS data from a WKY/N rat to both Rnor_6.0 and mRatBN7.2 and called variants against each respective genome. We then used the UCSC Liftover tool[85] and corresponding chain files to lift these variants to the other reference genome. The resulting lifted variant sets are then compared to the directly called variant sets.

### Identify live strains that are close to HS progenitors

Currently, there are two colonies of the HS rats: one located at Wake Forest University (RRID: RGD_13673907), which was moved from the Medical College of Wisconsin (RRID: RGD_2314009); the University of California San Diego houses the other colony (RID: RGD_155269102). While the original progenitor population is no longer alive, we obtained DNA samples that were preserved in 1984, when the HS colony was created. Using the 163-by-163 IBS matrix previously generated (see method: tree generation). We identified six closely-related substrains of the progenitors that were over 99.5% similar to the original strains based on identity by state (IBS): ACI/EurMcwi, BN/NHsdMcwi, F344/DuCrl, M520/NRrrcMcwi, MR/NRrrc WKY/NHsd. The best matches of the remaining strains were less similar: BUF/Mna (73.6%) for BUF/N and WAG/RijCrl (72.0%) for WN/N. Better alternatives for these two strains may be identified in the future as more inbred rat strains are sequenced. At the time of this report, all 8 of these inbred strains are available from Hybrid Rat Diversity Program at the Medical College of Wisconsin (see https://rgd.mcw.edu/wg/hrdp_panel/for strain availability and contact details).

### Constructing a genetic map using genetic data from a large HS cohort

The collection of genotypes from 1893 HS rats and 753 parents (378 families) was described previously.[15] Briefly, genotypes were determined using genotyping-by-sequencing.[95] This produced approximately 3.5 million SNP with an estimated error rate <1%. Variants for X- and Y chromosomes were not called. The genotype data used for this study can be accessed from the C-GORD database at https://doi.org/10.48810/P44W2 or through https://www.genenetwork.org. The genotype data were further cleaned to remove monomorphic SNPs. Genotypes with high Mendelian inheritance error rates (>2% error across the cohort) were identified by PLINK[86] and removed. We further used an unbiased selection procedure, which yielded a final list of 150,835 binned markers distributed across the genome. Mean distance between markers is 18.5 kb across the genome. We used Lep-MAP3 (LM3)[87] to construct the genetic map. The following LM3 functions were used: 1) the ParentCall2 function was used to call parental genotypes by taking into account the genotype information of grandparents, parents, and offspring; 2) the Filtering2 function was used to remove those markers with segregation distortion or those with missing data using the default setting of LM3; and 3) the OrderMarkers2 function

was used to compute cM distances (i.e., recombination rates) between all adjacent markers per chromosome. The resulting map had consistent marker order that supported the mRatBN7.2.

### Phylogenetic tree

We used bcftools[88] to filter for high-quality, bi-allelic SNP sites from the VCF files for the 20 autosomes. We then employed PLINK[86] to calculate a pairwise identity-by-state (IBS) matrix using the 11.5 M variants. We imported the resulting matrix into R and converted it to a meg format as described in the MEGA manual. We then used MEGA[89] to construct a distance-based UPGMA tree using the meg file as input. We used MEGA's "flip subtree" function to adjust the position of a few internal nodes for improved visualization using the MEGA GUI. The modified tree was exported as a nwk file. We then imported this nwk file into R and used the ggtree[90] package to plot the phylogenetic tree. There is a pdf file document the detailed steps of phylogenetics analysis available in key resources table "Code and data for phylogenetic analysis".

### RNA-seq data

RNA-seq data was downloaded from RatGTEx.[30] Brains were extracted from 88 HS rats. Rats were housed under standard laboratory conditions. Rat brains were extracted and cryosectioned into 60 μm sections, which were mounted onto RNase-free glass slides. Slides were stored in −80°C until dissection and before RNA-extraction post dissection. AllPrep DNA/RNA mini kit (Qiagen) was used to extract RNA. RNA-seq was performed on mRNA from each brain region using Illumina HiSeq 4000 to obtain 100 bp single-end reads for 435 samples. RNA-Seq reads were aligned to the Rnor_6.0 and mRatBN7.2 genomes from Ensembl using STAR v2.7.8a.[91]

### eQTL relocation analysis

We obtained the nucleus accumbens core (NAcc, 75 samples) eQTL dataset from RatGTEx[30] (https://ratgtex.org), which was mapped using Rnor_6.0. We considered associations with $p < 1 \times^{-8}$ between any observed SNP and any gene. We labeled those for which the SNP was within 1Mb of the gene's transcription start site as *cis*-eQTLs, and those with TSS distance greater than 5Mb, or with SNP and gene on different chromosomes, as *trans*-eQTLs. SNP-gene pairs with TSS distance 1-5Mb were not counted in either group. We estimated the set of cross-chromosome genome segment translocations between Rnor_6.0 and mRatBN7.2 using Minimap2[80] with the "asm5" setting. Examples of the relocations were visualized using the NCBI Comparative Genome Viewer (https://ncbi.nlm.nih.gov/genome/cgv/).

### Capped small (cs)RNA-seq data

csRNA-seq data used for the alignment metrics were previously published.[75] Briefly, small RNAs of ∼15–60 nt were size selected by denaturing gel electrophoresis starting from total RNA extracted from 14 rat brain tissue dissections. For csRNA libraries, cap selection was followed by decapping, adapter ligation, and sequencing. For input libraries, 10% of small RNA input was used for decapping, adapter ligation, and sequencing. After library quality check by gel electrophoresis, the samples were sequenced using the Illumina NextSeq 500 platform using 75 cycles single end. Sequencing reads were aligned to the Rnor_6.0 and rat mRatBN7.2 genome assembly using STAR v2.5.3a[91] aligner with default parameters. Transcriptional start regions were defined using HOMER's findPeaks tool.[92]

### Single nuclei (sn) RNA-seq data

snRNA-seq data used for the alignment metrics were obtained from rat amygdala using the Droplet-based Chromium Single-Cell 3′ solution (10x Genomics, v3 chemistry), as previously described.[96] Briefly, nuclei were isolated from frozen brain tissues and purified by flow cytometry. Sorted nuclei were counted and 12,000 were loaded onto a Chromium Controller (10x Genomics). Libraries were generated using the Chromium Single-Cell 3′ Library Construction Kit v3 (10x Genomics, 1000075) with the Chromium Single-Cell B Chip Kit (10x Genomics, 1000153) and the Chromium i7 Multiplex Kit for sample indexing (10x Genomics, 120262) according to manufacturer specifications. Final library concentration was assessed by Qubit dsDNA HS Assay Kit (Thermo-Fischer Scientific) and post library QC was performed using Tapestation High Sensitivity D1000 (Agilent) to ensure that fragment sizes were distributed as expected. Final libraries were sequenced using the NovaSeq6000 (Illumina). Sequencing reads were aligned to the Rnor_6.0 and rat mRatBN7.2 genome assembly using Cell Ranger 3.1.0.

### Transcriptome termini site sequencing

A total of 83 WTTS-seq (whole transcriptome termini site sequencing[29]) libraries were constructed individually using total RNA samples derived from several brain tissues of rats. Library sequencing produced a total of 312,092,803 raw reads, but the mapped reads were 240,225,046 (76.97%) on Rnor6.0, while 251,188,567 (80.49%) on mRatBN7.2, respectively. Using 25 reads per clustered site as a cutoff, we identified 173,124 APA sites mapped to the new reference genome, while 167,136 APA sites were assigned to the old reference genome. For Rnor6.0, only 127,460 (76.26%) APA sites were assigned to the genome regions with 18,177 annotated genes. In contrast, 141,399 (81.67%) APA sites were mapped to the 20,102 annotated genes on mRatBN7.2. In brief, our results provide evidence that mRatBN7.2 has improved qualities of both genome assembly and gene annotation in rats.

### Brain proteome data

Deep proteome data were generated using whole brain tissue from both parents and 29 members of the HXB family, one male and one female per strain. Proteins in these samples were identified and quantified using the tandem-mass-tag (TMT) labeling strategy coupled with two-dimensional liquid chromatography-tandem mass spectrometry (LC/LC-MS/MS). We used the QTLtools program[93] for protein expression quantitative trait locus (pQTL) mapping. *cis*-pQTL are defined when the transcriptional start sites for the tested protein are located within ±1 Mb of each other.

### Identifying potentially mislabeled samples

Phylogenetic analysis identified that the metadata of 15 samples contradicted their genetic relationships. These contradictions could happen at many steps during breeding, samples collection, sequencing, or data analysis and the true source is difficult to pinpoint. An advantage of having multiple biological samples for each strain is that these inconsistencies can be identified. One of the 15 samples is mislabeled and the correct label can be inferred (sample name denoted with ***); two samples are mislabeled and the correct labels cannot be inferred (sample names denoted with **); 12 samples are potentially mislabeled (sample names denoted with *). Details of these samples are described below.

(1) There is enough evidence to indicate that this sample is mislabeled, and we can conclusively infer what the true label is. One sample falls into this category: F344/Stm_HCJL.CRM. Despite being named F344, this sample has less than 70% IBS with the other 14 F344 samples, yet has about 99% IBS with 2 LE samples from different institutes (Table S10). We think this sample is mislabeled as F344, and the true label should be LE. Therefore, we changed the sample name accordingly and appended *** to the end of the sample name to denote such a change has been made.

(2) There is enough evidence to indicate that this sample is mislabeled, but we cannot conclusively infer what the true label is. Two samples fall into this category: LE/Stm_HCJL.CRM has about 83% IBS with the other 3 LE samples from different institutes (Table S10). The identity of this sample is likely one of the LEXF or FXLE recombinant inbred, but there isn't another sample that has high IBS with it. We think this sample is mislabeled, but the true label is unknown. WKY/Gla_TA.ILM is a sample we downloaded from SRA. This WKY substrain (WKY/Gla) clusters closer to SHR strains than other WKY substrains. The same pattern was also observed in one prior study,[41] and was thought to be caused by the incomplete inbreeding before sample distribution. To further investigate this, we performed regional similarity analysis and found that the pattern observed is consistent with that of a congenic strain created by using SHR as the recipient and WKY as the donor. A literature search confirmed that such strains were indeed once created at the same institute from where WKY/Gla was derived..[97] We think this sample is mislabeled, but we don't know what the correct label should be.

(3) There is evidence to suggest that this sample could potentially be mislabeled, but evidence is not conclusive. A total of 12 samples fall in this category. The majority of the samples of the same substrain but sequenced by different institutes have IBS over 99%; however, we observed a few instances of unexpected low IBS between samples of the same strain but with unexpected high IBS between samples of a different strain. These could be caused by the mis-labeling at either of the institutes. Although we think these samples are at the risk of being mislabeled, it is also possible that individual differences with these strains/substrains could be a cause of the unexpected IBS values. For example, both 7.7% of the variants of the BXH2 samples are heterozygous, while the rate of heterozygosity in the LEXF/FXLE in general is higher than the rest of the inbred strains. These pairs include (Table S10):

- BXH2_MD.ILM & BXH2_HCRW.CRM: unexpected low IBS at 92%
- LEXF4_MD.ILM & LEXF5_HCJL.CRM: unexpected high IBS at 99%
- LEXF3_MD.ILM & LEXF4_HCJL.CRM: unexpected high IBS at 99%
- LEXF1A_MD.ILM & LEXF1C_HCJL.CRM: unexpected high IBS at 99%
- LEXF1C_MD.ILM & LEXF2A_HCJL.CRM: unexpected high IBS at 99%
- LEXF2B_MD.ILM & LEXF1A_HCJL.CRM: unexpected high IBS at 98%
- LEXF4_MD.ILM & LEXF4_HCJL.CRM: unexpected low IBS at 83%
- LEXF1A_MD.ILM & LEXF1A_HCJL.CRM: unexpected low IBS at 84%
- LEXF1C_MD.ILM & LEXF1C_HCJL.CRM: unexpected low IBS at 95%

### Ensembl annotation

Annotation of the assembly was created via the Ensembl gene annotation system.[98] A set of potential transcripts was generated using multiple techniques: primarily through alignment of transcriptomic datasets, cDNA sequences, curated evidence, and also through gap filling with protein-to-genome alignments of a subset of mammalian proteins with experimental evidence from UniProt.[94] Additionally, a whole genome alignment was generated between the genome and the GRCm39 mouse reference genome using LastZ and the resulting alignment was used to map the coding regions of mouse genes from the GENCODE reference set.

The short-read RNA-seq data was retrieved from two publicly available projects; PRJEB6938, representing a wide range of different tissue samples such as liver or kidney, and PRJEB1924 which is aimed at understanding olfactory receptor genes. A subset of samples from a long-read sequencing project PRJNA517125 (SRR8487230, SRR8487231) were selected to provide high quality full length cDNAs.

From the 104 Ensembl release annotation on the rat assembly Rnor_6.0, we retrieved the sequences of manually annotated transcripts from the HAVANA manual annotation team. These were primarily clinically relevant transcripts, and represented high confidence cDNA sequences. Using the *Rattus norvegicus* taxonomy id 10116, cDNA sequences were downloaded from ENA and sequences with the accession prefix 'NM' from RefSeq.[76]

The UniProt mammalian proteins had experimental evidence for existence at the protein or transcript level (protein existence level 1 and 2).

At each locus, low quality transcript models were removed, and the data were collapsed and consolidated into a final gene model plus its associated non-redundant transcript set. When collapsing the data, priority was given to models derived from transcriptomic data, cDNA sequences and manually annotated sequences. For each putative transcript, the coverage of the longest open reading frame was assessed in relation to known vertebrate proteins, to help differentiate between true isoforms and fragments. In loci where the transcriptomic data were fragmented or missing, homology data was used to gap fill if a more complete cross-species alignment was available, with preference given to longer transcripts that had strong intron support from the short-read data.

Gene models were classified, based on the alignment quality of their supporting evidence, into three main types: protein-coding, pseudogene, and long non-coding RNA. Models with hits to known proteins, and few structural abnormalities (i.e., they had canonical splice sites, introns passing a minimum size threshold, low level of repeat coverage) were classified as protein-coding. Models with hits to known protein, but having multiple issues in their underlying structure, were classified as pseudogenes. Single-exon models with a corresponding multi-exon copy elsewhere in the genome were classified as processed pseudogenes.

If a model failed to meet the criteria of any of the previously described categories, did not overlap a protein-coding gene, and had been constructed from transcriptomic data then it was considered as a potential lncRNA. Potential lncRNAs were filtered to remove transcripts that did not have at least two valid splice sites or cover 1000bp (to remove transcriptional noise).

A separate pipeline was run to annotate small non-coding genes. miRNAs were annotated via a BLAST[99] of miRbase[100] against the genome, before passing the results into RNAfold.[101] Poor quality and repeat-ridden alignments were discarded. Other types of small non-coding genes were annotated by scanning Rfam[102] against the genome and passing the results into Infernal.[103]

The annotation for the rat assembly was made available as part of Ensembl release 105.

### RefSeq annotation

Annotation of the mRatBN7.2 assembly was generated for NCBI's RefSeq dataset[76] using NCBI's Eukaryotic Genome Annotation Pipeline.[104] The annotation, referred to as NCBI *Rattus norvegicus* Annotation Release 108, includes gene models from curated and computational sources for protein-coding and non-coding genes and pseudogenes, and is available from NCBI's genome FTP site and web resources.

Most protein-coding genes and some non-coding genes are represented by at least one known RefSeq transcript, labeled by the method "BestRefSeq" and assigned a transcript accession starting with NM_ or NR_, and corresponding RefSeq proteins designated with NP_ accessions. These are predominantly based on rat mRNAs subject to manual and automated curation by the RefSeq team for over 20 years, including automated quality analyses and comparisons to the Rnor_6.0 and mRatBN7.2 assemblies to refine the annotations. Nearly 80% of the protein-coding genes in AR108 include at least one NM_ RefSeq transcript, of which 33% have been fully reviewed by RefSeq curators.

Additional gene, transcript, and protein models were predicted using NCBI's Gnomon algorithm using alignments of transcripts, proteins, and RNA-seq data as evidence. The evidence datasets used for Release 108 are described at https://www.ncbi.nlm.nih.gov/genome/annotation_euk/Rattus_norvegicus/108/, and included alignments of available rat mRNAs and ESTs, 10.7 billion RNA-seq reads from 303 SRA runs from a wide range of samples, 1 million Oxford Nanopore or PacBio transcript reads from 5 SRA runs, and known RefSeq proteins from human, mouse, and rat. BestRefSeq and Gnomon models were combined to generate the final annotation, compared to the previous Release 106 annotation of Rnor_6.0 to retain GeneID, transcript, and protein accessions for equivalent annotations, and compared to the RefSeq annotation of human GRCh38 to identify orthologous genes. Gene nomenclature was based on data from RGD, curated names, and human orthologs.

