## [Document S2. Transparent peer review records for Tristan V. de Jong et al · Cell Genomics]

A revamped rat reference genome improves the discovery of genetic diversity in laboratory rats

Tristan V de Jong, 1† Yanchao Pan, 2† Pasi Rastas, 3 Daniel Munro, 4,5 Monika Tutaj, 6,7 Huda Akil, 8 Chris Benner, 9 Denghui Chen, 4 Apurva S Chitre, 4 William Chow, 10 Vincenza Colonna, 11,12 Clifton L Dalgard, 13 Wendy M Demos, 6,7 Peter A Doris, 14 Erik Garrison, 12 Aron M Geurts, 6 Hakan M Gunturkun, 1 Victor Guryev, 15 Thibaut Hourlier, 16 Kerstin Howe, 10 Jun Huang, 1 Ted Kalbfleisch, 17 Panjun Kim, 12 Ling Li, 12,18 Spencer Mahaffey, 19 Fergal J Martin, 16 Pejman Mohammadi, 20,21 Ayse Bilge Ozel, 2 Oksana Poleskaya, 4 Michal Pravenec, 22 Pjotr Prins, 12 Jonathan Sebat, 4 Jennifer R Smith, 6,7 Leah C Solberg Woods, 23 Boris Tabakoff, 19 Alan Tracey, 10 Marcela Uliano-Silva, 10 Flavia Villani, 12 Hongyang Wang, 24 Burt M Sharp, 12 Francesca Telese, 4 Zhihua Jiang, 24 Laura Saba, 19 Xusheng Wang, 12,18 Terence D Murphy, 25 Abraham A Palmer, 4,26 Anne E Kwitek, 6,7 Melinda R Dwinell, 6,7 Robert W Williams, 12 Jun Z Li, 2‡ Hao Chen 1‡

Summary

Initial submission: Received : 10/2/2023

Scientific editor: Judith Nicholson and Laura Zahn

First round of review: Number of reviewers: 2
Revision invited : 11/30/2023
Revision received : 12/26/2023

Second round of review: Number of reviewers: N/A
Accepted : February 29, 2024

Data freely available: Yes

Code freely available: Yes

This transparent peer review record is not systematically proofread, type-set, or edited. Special characters, formatting, and equations may fail to render properly. Standard procedural text within the editor's letters has been deleted for the sake of brevity, but all official correspondence specific to the manuscript has been preserved.

Referees' reports, first round of review

Reviewer #1:

The authors provide an analysis of the latest version of the Norway rat genome in a well written and conceived manuscript. I have only very minor comments.

"The Rattus norvegicus was sequenced shortly after the genomes of Homo sapiens and Mus musculus" should be "The Rattus norvegicus genome was sequenced..."

"Code availability - The code for the custom R, Python and Bash scripts for data analysis is available upon request." - I think this diminishes the usefulness of the manuscript for a greater genomics community. If the paper is meant to be useful for more than rat researchers, then the authors should consider a gitHub repository or a Zenodo submission with a descriptive README file that explains the scripts, dependencies, and their functions.

"Genetic relationships between strains in the RI panels can provide insights strain selection is needed when designing studies" - this sentence in the discussion needs to be edited or removed.

Reviewer #2: Hao Chen and colleagues provide an updated reference map of the Rat based on analysis of 163 short-read whole sequence datasets from 120 strains. Overall, this is the 7th iteration of the rat genome and an important contribution to the field. I am in full support of its publication in Cell Genomics

My suggestions are minor, in order to improve presentation.

1. Figure 8 is referred to as figure 6 in the text. The order should be changed, since figure 8 is discussed before figure 7.
2. The supplementary data are important, and need to be well presented.
 - a. supplementary figures 7-18 don't have a reference (figure number).
 - b. the legend is missing from most of the supplementary figures.
 - c. the fonts are small and hard to read in many of the figures (main and supplement).
 - d. general laxness in the presentation (figure S10 for instance has A, B, C.. randomly placed on the figure)

Authors' response to the first round of review

Reviewer #1:

Comment on the sentence: "The *Rattus norvegicus* was sequenced shortly after the genomes of *Homo sapiens* and *Mus musculus*."

We have revised this sentence to "The *Rattus norvegicus* genome was sequenced shortly after the genomes of *Homo sapiens* and *Mus musculus*."

Code and data availability: We appreciate the emphasis on broader utility and accessibility. We have uploaded our code to a GitHub repository: <https://github.com/hanyoupan/rat-manuscript>. We have added several detailed README files describing the scripts. In addition, we have uploaded additional resources to Zenodo and provided their DOI in the additional resource table. We believe these code and data will significantly enhance the manuscript's utility for the genomics community.

Comments on the sentence on "Genetic relationships between strains in the RI panels can provide insights strain selection is needed when designing studies." Upon review, we agree that the sentence mentioned is unclear and unnecessary. We have removed this sentence.

Reviewer #2:

Inconsistency in Figure References: We acknowledge the error in referring to Figure 8 as Figure 6. This has been corrected to ensure consistency and clarity in the text.

Presentation of Supplementary Data:

- a. Reference to Supplementary Figures: We have cited all supplementary figures (7-18) in the main text.
 - b. Legend for Supplementary Figures: All supplementary figures now include detailed legends.
 - c. Font Size in Figures: We adjusted the font sizes in figures (main and supplementary) to enhance readability.
 - d. Overall Presentation Quality: We have reviewed and revised the presentation of all figures, particularly Figure S10, to ensure a professional and coherent layout.
-